# Comparison of the Diagnostic Performance of Deep Learning Algorithms for Reducing the Time Required for COVID-19 RT–PCR Testing

**DOI:** 10.3390/v15020304

**Published:** 2023-01-22

**Authors:** Yoonje Lee, Yu-Seop Kim, Da In Lee, Seri Jeong, Gu Hyun Kang, Yong Soo Jang, Wonhee Kim, Hyun Young Choi, Jae Guk Kim

**Affiliations:** 1Department of Emergency Medicine, College of Medicine, Hallym University, Chuncheon 24252, Republic of Korea; 2Hallym Bioinformatics & Convergence Research Laboratory, Hallym Translation Research Center, Kangnam Sacred-Heart Hospital, Hallym University, Seoul 07441, Republic of Korea; 3Department of Convergence Software, Hallym University, Chuncheon 24252, Republic of Korea; 4Department of Laboratory Medicine, Kangnam Sacred Heart Hospital, College of Medicine, Hallym University, Seoul 07441, Republic of Korea

**Keywords:** SARS-CoV-2, COVID-19, deep learning, RT-PCR, diagnosis

## Abstract

(1) Background: Rapid and accurate negative discrimination enables efficient management of scarce isolated bed resources and adequate patient accommodation in the majority of areas experiencing an explosion of confirmed cases due to Omicron mutations. Until now, methods for artificial intelligence or deep learning to replace time-consuming RT-PCR have relied on CXR, chest CT, blood test results, or clinical information. (2) Methods: We proposed and compared five different types of deep learning algorithms (RNN, LSTM, Bi-LSTM, GRU, and transformer) for reducing the time required for RT-PCR diagnosis by learning the change in fluorescence value derived over time during the RT-PCR process. (3) Results: Among the five deep learning algorithms capable of training time series data, Bi-LSTM and GRU were shown to be able to decrease the time required for RT–PCR diagnosis by half or by 25% without significantly impairing the diagnostic performance of the COVID-19 RT–PCR test. (4) Conclusions: The diagnostic performance of the model developed in this study when 40 cycles of RT–PCR are used for diagnosis shows the possibility of nearly halving the time required for RT–PCR diagnosis.

## 1. Introduction

Until the emergence of the Omicron mutation, most nations have tried to implement strategies to quickly find positive cases, isolate them, initiate early treatment, and precisely identify negative patients to minimize the spread of infection and to avoid advancement to a critical disease condition.

Nevertheless, with the emergence of Omicron mutations, SARS-CoV-2 infection, which began in the winter of 2019 and has spread throughout the world, entered a new phase. The emergence of Omicron has resulted in a rapid increase in the number of confirmed cases, and as a result of this rising trend, many countries around the world have been exposed to a situation that is difficult to manage with the current medical capabilities associated with COVID-19. In this situation, the previously employed strategy of the rapid isolation of patients, the confirmation of the patient’s diagnosis and treatment, and the tracking of the patient’s contacts is no longer feasible. Although RT–PCR is still considered the gold standard for the confirmation of a COVID-19 diagnosis because of the high diagnostic accuracy of RT–PCR, this test is labour-intensive and takes a longer time than the rapid antigen test (RAT). Furthermore, countries in this situation are contemplating the use of or are currently employing the RAT rather than RT–PCR testing as an alternative method.

As crucial as it is to precisely detect any positive cases, the accurate and rapid detection of negative results is still critical for preventing the spread of infection. From this perspective, it is critical to evaluate possible cases with the RAT in this situation.

The RAT has the advantage of a short test time and high diagnostic accuracy when performed on symptomatic people or on people with a high viral load, but its diagnostic accuracy is low for asymptomatic people [1,2,3,4,5,6,7]. Its negative predictive value is also insufficient, which lowers the performance to detect true negatives. Because the dependability of negative results, i.e., the negative predictive value, is insufficient, there is a possibility that there will be an actual infected individual among those patients who had tested negative on the RAT. As several previous studies have evidenced, in the current situation, when the prevalence is rising, the RAT is not seen as an acceptable method to reduce the risk of infectious disease propagation [1,2,3,4,5,6,7].

Every country is fully aware of such issues; however, the adoption of the RAT is unavoidable in the current situation because the spread of this infectious disease is constantly expanding.

Through a previous study, we recently developed and evaluated a deep learning algorithm that can reduce COVID-19 RT–PCR diagnostic time [8]. In the previous study, only the long short-term memory (LSTM) deep learning (DL) algorithm, which is known to be suitable for time series data, was used.

However, in the current study, five DL algorithms (① recurrent neural network, RNN; ② long short-term Memory, LSTM; ③ bidirectional long short-term memory, BiLSTM; ④ gait recurrent unit, GRU; ⑤ transformer) [9,10,11,12,13], including LSTM, known to be suitable for time series data were implemented by utilizing more data. We then compared the diagnostic performance of each algorithm and evaluated its clinical applicability.

### Related Works

Since the COVID-19 pandemic, numerous attempts have been made to identify COVID-19 early using technologies other than RT-PCR, including artificial intelligence and deep learning. However, these earlier investigations utilized X-ray or CT images, blood test results, or vital signs gathered from patients as diagnosis tools, and did not produce data that could totally substitute RT-PCR [16,17,18,19,20,21,22,23,24,25,26,27,28,29,30,31,32,33,34].

## 2. Materials and Methods

### 2.1. Study Participants

We enrolled patients who visited a specialized outpatient department for COVID-19 or who visited an emergency department for possible COVID-19 between 23 November 2020 and 25 September 2021. During this period, a total of 27,835 raw RT–PCR curve data points were obtained to identify cases of COVID-19. For this research, 1270 positive and 1270 negative test results were chosen from among these cases.

This study was approved by the Institutional Review Committee (HKS 2020-07-007) of Hallym University Kangnam Sacred Heart Hospital in Korea; the requirement for informed consent was waived because the subjects’ data were anonymized. This study was conducted in accordance with the STARD guidelines and regulations for a study related to the diagnostic accuracy of COVID-19 RT–PCR.

### 2.2. Materials

A MagNa Pure 96 System was used to extract RNA from the samples (Roche Diagnostics, Rotkreuz, Switzerland). A STANDARD M nCoV Real-Time Detection kit (SD Biosensor, Gyeonggi, Republic of Korea) was utilized in this investigation, and a Bio–Rad CFX96 analyser (Bio–Rad Laboratories, Inc., Hercules, CA, USA) was used for the RT–PCR test (Figure 1).

### 2.3. Data Description

The RT–PCR results from nasopharyngeal swab specimens of the patients who received SARS-CoV-2 RT–PCR testing at Kangnam Sacred Heart Hospital were included in the raw data. The fluorescence values that were measured for a total of 40 cycles via the RT–PCR test were recorded for each patient sample, and the raw data comprised the fluorescence values that were produced during the RT–PCR testing for SARS-CoV-2.

As a result, for each sample, 40 fluorescence values were measured across 40 cycles, yielding a total of 2540 raw data points. The fluorescence readings were recorded in 40 columns for each sample across a total of 2540 rows in the raw data. There were 1270 verified positive and 1270 verified negative test results in total.

### 2.4. Development of the DL Model

The output variable for training the models was the RT–PCR findings (positive or negative). A total of 40 models were produced and evaluated, starting with the model trained with only the fluorescence value of the first RT–PCR cycle, and the last model was the model trained with the fluorescence value of all 40 RT–PCR cycles.

The first model, for example, was trained using the fluorescence value from the first RT–PCR cycle, whereas the second model was trained using the fluorescence values from the first and the second RT–PCR cycles. Similarly, the fluorescence values from the first to the 39th RT–PCR cycle were used to train the 39th model, and the fluorescence values from the first to the 40th RT–PCR cycle were used to train the 40th model.

The raw data for the RT–PCR fluorescence values that were used in this study exhibited sequential characteristics that varied with the extraction time during the cycle. Thus, the different deep learning models (“Recurrent Neural Network” (RNN), “Long Short-Term Memory” (LSTM), “Bidirectional Long Short-Term Memory” (Bi-LSTM), “Gated Recurrent Unit” (GRU), and “Transformer”) that were suitable for time series processing were applied in this study using Tensorflow (Hardware: GPU; RTX 2080 Ti * 2, CPU; i7-9800X, RAM; 64GB).

An RNN is a simple recurrent neural network that sequentially processes time series data [9]. LSTM is a neural network used to prevent vanishing gradients, which are known to be a problem with existing RNNs [10]. This model has a four-layer structure and consists of three gates (forget, input, output). The forget gate determines how much past information is to be forgotten, and the input gate determines how much current information is to be remembered. The output gate is a gate for exporting the final hidden result. The existing LSTM learns in the forward direction, but the BiLSTM also learns in the reverse direction [11]. Therefore, both outputs from bidirectional learning are used for prediction. The GRU has a simplified model structure compared to the existing LSTM. The GRU consists of two gates (update, reset), and it solves the long-term dependency problem and reduces computation. The reset gate decides how to merge the new input with the old memory, and the update gate decides how much to keep of the old memory [12]. Transformer models, on the other hand, were originally comprised of encoders and decoders and are typically employed for sequence-to-sequence learning tasks, such as translation [13]. Encoder blocks comprised of normalization and attention were employed in this investigation. Instead of learning sequentially, the encoder learned through attention weights (Figure 2). The information on the optimal hyper-parameters of DL models based on the grid search method is provided in Table 1, and we used the early stopping function to prevent the problem of over-fitting.

Through these proposed models, the positive and negative RT–PCR results were classified, and then the results were compared and analysed.

### 2.5. Training and Test Datasets

To train the models, the results of the RT–PCR virology tests were utilized as a reference. There were 1270 positive RT–PCR results among the 2540 patients whose data were included in the research, and 1270 of the patients had negative results. These data were split into training (2000) and test (540) datasets. Raw data from curves of the RT–PCR results from 1000 positive and 1000 negative cases were used to establish the data for the training and validation in an 80:20 ratio. For testing, 270 positive and 270 negative results were utilized (Figure 3).

### 2.6. Outcomes

The primary endpoints were the sensitivity, specificity, AUROC values, positive predictive value (PPV), negative predictive value (NPV), and accuracy. The PPV, NPV and accuracy were assessed using a 5% prevalence assumption and in consideration of the rapid surge in the number of confirmed cases of the Omicron variant.

The secondary endpoints were the comparisons of the diagnostic performances in each of the algorithms of models 10 and 20. 

### 2.7. Statistical Analysis

SPSS software V.26.0 was used for all statistical analyses (IBM, SPSS, Inc., Chicago, IL, USA). In addition to the positivity or negativity of RT–PCR data, the sensitivity (the proportion of actual positives) and specificity (the proportion of actual negatives) were also determined. The mean and 95% confidence interval are used to express the variables. To compare diagnostic performance between algorithms, the AUROC values of each were compared to those of the DeLong method; statistical significance was defined as a *p* value less than 0.05. (*p* < 0.05).

## 3. Results

### 3.1. The Diagnostic Performance of Each DL Algorithm

The sensitivities of the algorithm (Model No. 10) that was trained with the raw data up to 10 cycles (which was the first 25% of the total of 40 cycles) were 96.7 (95% CI, 93.8–98.2) for LSTM and Bi-LSTM, 95.6 (95% CI, 92.4–97.4) for GRU, 90.4 (95% CI, 86.3–93.3) for RNN, and 88.5 (95% CI, 84.2–91.8) for transformer. The sensitivities in the algorithm (Model No. 20) that was trained from the raw data up to 20 cycles (which was half of the entire total number of cycles) were 97.8 (95% CI, 95.2–99.0) for LSTM, 97.4 (95% CI, 94.8–98.7) for GRU, 97.0 (95% CI, 94.3–98.5) for Bi-LSTM and RNN, and 94.4 (95% CI, 91.0–96.6) for transformer (Figure 4A, Appendix A).

In model No. 10, the specificities were 73.7 (95% CI, 68.2–78.6) for Bi-LSTM, 73.0 (95% CI, 67.4–77.9) for GRU, 69.6 (95% CI, 63.9–74.8) for LSTM, 68.5 (95% CI, 62.8–73.8) for RNN, and 67.8 (95% CI, 62.0–73.1) for transformer. In model No. 20, the specificities were 89.3 (95% CI, 85.0–92.4) for Bi-LSTM and transformer, 87.4 (95% CI, 82.9–90.9) for GRU, 86.7 (95% CI, 82.1–90.2) for RNN, and 86.3 (95% CI, 81.7–89.9) for LSTM (Figure 4B, Appendix A).

The areas under the receiver operating characteristic (AUROC) were 85.2 (95% CI, 82.2–88.1) for Bi-LSTM, 84.3 (95% CI, 81.6–87.1) for GRU, 83.2 (95% CI, 80.3–85.9) for LSTM, 79.4 (95% CI, 76.1–82.7) for RNN, and 78.2 (95% CI, 74.9–81.7) for transformer, respectively, in model No. 10; in model No. 20, they were 93.2 (95% CI, 91.0–95.0) for Bi-LSTM, 92.4 (95% CI, 90.3–94.5) for GRU, 92.0 (95% CI, 89.9–94.2) for LSTM, 91.9 (95% CI, 89.7–94.0) for RNN, and 91.9 (95% CI, 89.5–94.0) for transformer (Figure 4C, Appendix A).

### 3.2. The Effect of the Prevalence on the Diagnostic Performance of Each DL Algorithm

As of 25 February 2022, the calculated prevalence rates in Korea, Germany, and the United States were 2.96 percent, 4.49 percent, and 8.42 percent, respectively. Considering the trend in the current prevalence, which has been affected by the Omicron mutation, a prevalence of 5% was utilized to calculate the PPV and NPV affected by the prevalence in this study [14].

In model No. 10, the PPVs were 16.2 (95% CI, 13.7–19.1) for Bi-LSTM, 15.7 (95% CI, 13.2–18.5) for GRU, 14.4 (95% CI, 12.3–16.7) for LSTM, 13.1 (95% CI, 11.2–15.3) for RNN, and 12.6 (95% CI, 10.8–14.7) for transformer. In model No. 20, the PPVs were 32.2 (95% CI, 25.2–40.2) for Bi-LSTM, 31.6 (95% CI, 24.7–39.5) for transformer, 28.9 (95% CI, 22.9–35.8) for GRU, 27.7 (95% CI, 22.0–34.2) for RNN, and 27.3 (95% CI, 21.8–33.6) for LSTM (Figure 5A, Appendix A).

In model No. 10, the NPVs were 99.8 (95% CI, 99.6–99.9) for Bi-LSTM, 99.8 (95% CI, 99.5–99.9) for LSTM, 99.7 (95% CI, 99.4–99.8) for GRU, 99.3 (95% CI, 98.9–99.5) for RNN, and 99.1 (95% CI, 98.8–99.4) for transformer. In addition, in model No. 20, the NPVs were 99.9 (95% CI, 99.7–99.9) for LSTM, 99.8 (95% CI, 99.7–99.9) for Bi-LSTM, 99.8 (95% CI, 99.7–99.9) for GRU, 99.8 (95% CI, 99.6–99.9) for RNN, and 99.7 (95% CI, 99.5–99.8) for transformer (Figure 5B, Appendix A).

The accuracies in model No. 10 were 74.9 (95% CI, 70.9–78.5) for Bi-LSTM, 74.1 (95% CI, 70.2–77.7) for GRU, 70.9 (95% CI, 66.9–74.8) for LSTM, 69.6 (95% CI, 65.5–73.5) for RNN, and 68.8 (95% CI, 64.7–72.7) for transformer, respectively; in model No. 20, the accuracies were 89.7 (95% CI, 86.8–92.1) for Bi-LSTM, 89.5 (95% CI, 86.6–91.9) for transformer, 87.9 (95% CI, 84.9–90.5) for GRU, 87.2 (95% CI, 84.1–89.9) for RNN, and 86.9 (95% CI, 83.7–89.6) for LSTM (Figure 5C, Appendix A).

### 3.3. Comparison of the Diagnostic Performance of the Algorithms in Models 10 and 20

The AUROCs appeared as follows in model No. 10 according to each algorithm. The AUROC for Bi-LSTM was 85.2 (95% CI, 82.2–88.1), the AUROC for GRU was 84.3 (95% CI, 81.6–87.1), the AUROC for LSTM was 83.2 (95% CI, 80.3–85.9), the AUROC for RNN was 79.4 (95% CI, 76.1–82.2), and the AUROC for transformer was 78.2 (95% CI, 74.9–81.7). There were no statistically significant differences (*p* > 0.05) between “RNN vs. transformer”, “LSTM vs. GRU”, and “Bi-LSTM vs. GRU” (Figure 6A, Table 2).

In Model No. 20, the AUROC values were 93.2 (95% CI, 91.0–95.0) for Bi-LSTM, 92.4 (95% CI, 90.3–94.5) for GRU, 92.0 (95% CI, 89.9–94.2) for LSTM, 91.9 (95% CI, 89.7–94.0) for RNN, and 91.9 (95% CI, 89.5–94.0) for transformer. In a pairwise comparison, there was no statistically significant difference in the AUROC values of the algorithms in Model No 20 (all *p* > 0.1) (Figure 6B, Table 3).

## 4. Discussion

Prior to the discussion, it is vital to clarify the objective of this study and the significance of the result values that are displayed in the study findings to properly convey the study’s meaning. This study aims to develop and validate models for reducing the RT–PCR test time by utilizing five deep learning methods for time-domain data processing. Therefore, in the results of this study, we confirmed and compared the performance of several models. “Model No. 20 learned by using raw data up to 20 cycles and can decrease the RT–PCR test time by 50%”, and “Model No. 10 learned using the raw data up to 10 cycles and can decrease the test time in by 25%”.

In model No. 10, the AUROCs when using Bi-LSTM and GRU were 85.2 (82.2–88.1) and 84.3 (81.6–87.1), respectively, and this model demonstrated the best diagnostic performance. No statistically significant difference existed between the two algorithms. These two algorithms significantly outperformed the rest of the algorithms. There was no statistically significant difference between all of the algorithms in Model No. 20; however, the algorithm using Bi-LSTM had an AUROC of 93.2 (91.0–95.0), which demonstrated that this model had the best diagnostic performance. Additionally, Bi-LSTM demonstrated superior performance not only in terms of sensitivity and specificity but also in terms of PPV, NPV, and accuracy, which are influenced by prevalence.

According to the literature, the diagnostic performance was evaluated as no discrimination when the AUROC value was less than 50, acceptable when the AUROC value was between 70 and 80, excellent when the AUROC value was between 80 and 90, and outstanding discrimination when the AUROC value was greater than 90 [14,15,16,17]. The diagnostic performances of the models that were developed in this study were close to or exceeded 80 in Model No. 10 regardless of the algorithm, which indicates that there was generally excellent discrimination, and the diagnostic performance exceeded 90 in Model No. 20 regardless of the algorithm, indicating outstanding discrimination.

In terms of sensitivity, Kim et al. [18]. previously reported that the pooled sensitivity of RT–PCR was 89.0 (95% CI, 81.0–94.0), and an earlier meta-analysis study by Hayer et al. [19]. indicated that the overall sensitivity of RAT was 74.7 (95% CI, 63.7–80.9). However, Ricco et al. [20] indicated that the pooled sensitivity of RAT was 64.8 (95% CI, 54.5–74.0). In this study, it was confirmed that the sensitivities of all the algorithms of Model No. 10 exceeded 90.

The sensitivities calculated in this study tended to be slightly lower compared to Ricco et al.’s [20] meta-analysis, which found that the pooled specificity of RAT was 98.0 (95% CI, 95.8–99.0). The sensitivity in Model No. 10 was close to or exceeded 70, and in Model No. 20, it was close to 90. However, for the specificity to be clinically meaningful as a means of preventing the spread of infection, the NPV, which changes according to prevalence, must be sufficiently high. However, the specificities of the models that were not impacted by the prevalence were slightly lower than that of the RAT in this investigation. However, the NPVs in all of the algorithms of Model No. 10 and No. 20 that were produced in this study were greater than 99.0, compared to a previous meta-analysis study by Ricco et al. [20], which found a range of 26.2 to 94.1. In other words, there is a 99% or greater likelihood that a verified negative patient is truly negative, suggesting that these algorithms may be safe screening approaches for preventing viral spread.

The PPVs (with a prevalence of 5%) in this study were observed to have a range of 12.6 to 16.2 in Model No. 10 and a range of 27.3 to 32.2 in Model No. 20. The PPVs have ranged from 57.1 to 100.0 in studies that did not reflect the difference in the prevalence, and this was previously demonstrated in a meta-analysis study on the RAT diagnostic performance by Ricco et al. [20].

Studies regarding various artificial intelligence, deep learning, and machine learning methods to replace RT–PCR, which is the gold standard for the diagnosis of COVID-19, have been reported.

Studies that have evaluated the COVID-19 diagnostic performance of models trained with CT imaging data using the convolutional neural network (CNN) approach reported sensitivities of 77.0 to 90.0, specificities of 68.0 to 96.6, and AUROCs of 0.85 to 0.97 [21,22,23,24,25,26,27,28,29]. In the same way, studies that evaluated the diagnostic performances of models that were trained using chest X-ray images previously reported sensitivities of 78.0 to 97.0, specificities of 72.6 to 99.2, and AUROCs of 0.77 to 0.92 [30,31,32,33,34,35,36]. The diagnostic performance of models that were trained with clinical information and some blood test results using the RNN method have been previously reported with sensitivities of 66.0 to 93.0, specificities of 64.0 to 97.9, and AUROCs of 0.86 to 0.98 [37,38,39].

As mentioned previously and in view of the results of various previous studies that have evaluated the replacement of the RT–PCR diagnostic test, the findings of these previous studies seem to be applicable to the clinical setting, and the goal is to minimize the possibility for viral transmission through the quick identification of patients, the isolation of patients, and the safe discharge of patients from isolation.

Nonetheless, it is challenging to utilize these diagnostics in a clinical setting due to the following issues.

These previous studies have indicated that there are problems, including imbalance and bias, in the data used for the model learning because of the study limitations [18,19,20,21,22,23,24,25,26,27,28,29,30,31,32,33,34,35,36,37,38,39]. Laghi concurs with efforts to diagnose COVID-19 using AI models [40]. However, given the clinical course of SARS-CoV-2 infection, the models provided in these studies were not developed by reflecting the difference in the test results across the time frame from the onset of infection to test execution. As a result, the use of these models in a true clinical setting appears to be extremely risky.

Considering this point, the prior study had a bias in the data used for learning, and it assessed a model that only utilized a single algorithm, LSTM. However, in the present study, 1270 negative results and 1270 positive results were acquired and applied at the same rate to boost the reliability of the learned models. Additionally, by learning through five different algorithms, it was feasible to evaluate each algorithm’s diagnostic performance.

In this study, the results of sensitivity, specificity, and AUROC show similarly high performance in LSTM, GRU, and Bi-LSTM. This is thought to show the tendency of RNN-based models that show excellent performance in processing time series data. In other words, through this study, it was confirmed that RNN-type models are very suitable deep learning models for training RT-PCR time series data. 

In contrast, RNNs and transformers demonstrated poor performance. The following are the reasons for these results: The gradient vanishing problem is a limitation of RNN in which previous information is lost as the length of the time series increases, and as the cycle lengthens, initial information is lost, resulting in poor performance. Additionally, the transformer’s structure is more complex than other DL models. Consequently, it requires a large quantity of data to be utilized effectively, but the data in this study are relatively small compared to the characteristics of the domain, so it can be interpreted as demonstrating poor performance. 

Therefore, RT-PCR data were deemed unsuitable for transformers, which are extremely complex and large models, or very simple RNN models, and it was determined that LSTM, GRU, and Bi-LSTM are more applicable DL models for clinical applications. In the future, we expect to further improve the generalization performance of the DL model by collecting more data and redesigning the structure of the DL model, so that it can be effectively used in the clinical field.

Nevertheless, this study has several limitations as well. First, the outcomes of this study do not represent differences in race or geographical area, as the data that were used to train and test each algorithm-specific model were obtained in South Korea. However, the fluorescence values that were acquired during the 40 cycles of RT–PCR are the same regardless of race or region, and the RT–PCR test procedure is identical regardless of the geographical region or race. As a result, even if there are differences in the results due to the differences in race or geographical area, there is no need to collect further data to reflect these differences for the model learning and testing. Second, in the case of the PPV, which is one of the variables that is affected by the prevalence, the performance of the models that were developed in this work is poor in comparison to the RT–PCR or RAT PPV values. However, when the previous studies examined the diagnostic performance of RT–PCR or RAT, either the prevalence was not addressed in those studies or the PPV was determined in those studies based on diagnostic data that were obtained from certain population groups, such as symptomatic patients. As a result, the values may appear inflated. Thus, it is difficult to draw direct comparisons between the findings in those studies and the findings of this study. Third, because the patient’s symptoms, blood test results, and X-ray test results were not linked to the data that were used for learning and testing in this study, these additional data were not used in conjunction with the data used for learning and testing; therefore, it is unreasonable to apply these models directly in the clinical field. If the patient’s sex, age, symptoms, vital signs, blood test results, or X-ray test findings are all analysed and used for learning, it is believed that it will be feasible to determine whether the model is beneficial and can be applied to real-world patients. However, the models in this study were not constructed in this manner. Nonetheless, to our knowledge, except for the previous study that we conducted, no study has been undertaken on the diagnostic performance of deep learning models trained for the goal of lowering the time necessary for RT–PCR diagnosis by utilizing raw data of fluorescence values from 40 cycles of RT–PCR. If the models that were developed in this study are combined with other clinical data in the future, it is possible that a diagnostic approach for a variety of infectious diseases will be conceivable; therefore, additional research will be required in the future.

## 5. Conclusions

Among the five deep learning algorithms capable of training time series data, Bi-LSTM and GRU were shown to be suitable for halving or quartering the time required for RT–PCR diagnosis without significantly impairing the diagnostic performance of the COVID-19 RT–PCR test.

## Figures and Tables

**Figure 1 viruses-15-00304-f001:**
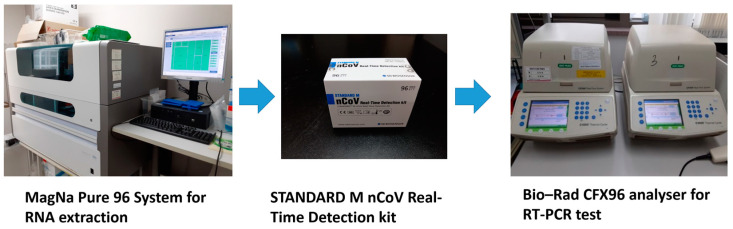
Equipment and reagent used for RT-PCR diagnostic test.

**Figure 2 viruses-15-00304-f002:**
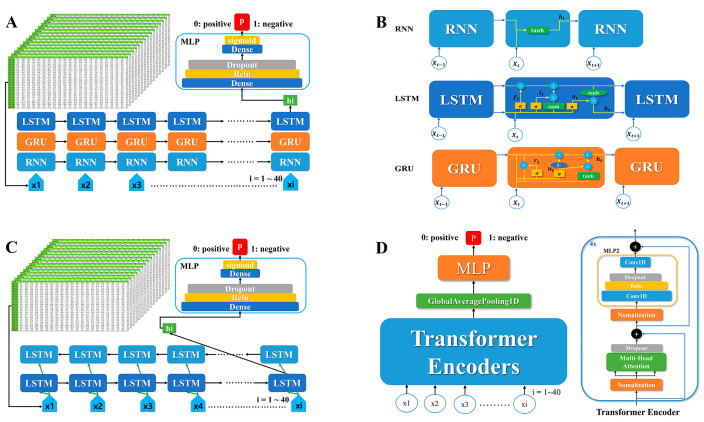
Architectures of five different DL algorithms. (**A**) General architectures of DL algorithms for RNNs, LSTM and GRUs. (**B**) Details of the architectures for each DL algorithm for RNN, LSTM and GRU. (**C**) Details of the architecture of the Bi-LSTM algorithm. (**D**) Details of the architecture of the transformer algorithm. RNN: recurrent neural network; LSTM: long short-term memory; GRU: gated recurrent unit; Bi-LSTM: bidirectional long short-term memory; DL: deep learning.

**Figure 3 viruses-15-00304-f003:**
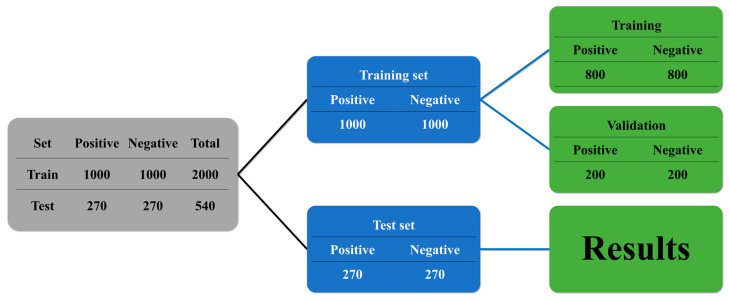
Composition of the training and test datasets.

**Figure 4 viruses-15-00304-f004:**
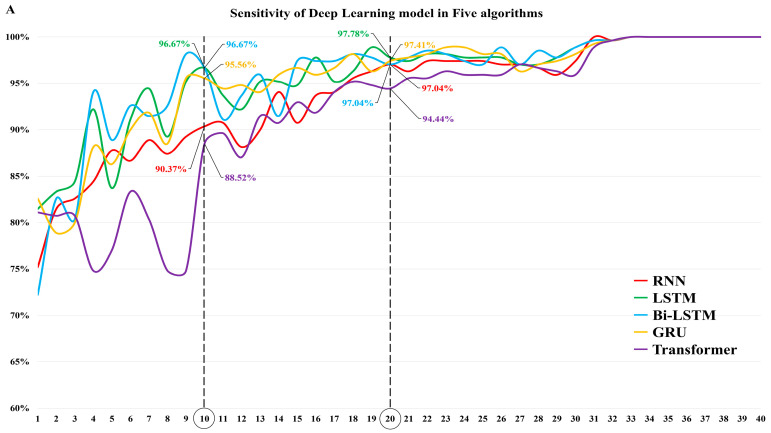
Diagnostic performance of DL models in five different algorithms using the test dataset. (**A**) Sensitivity of DL model in five algorithms. (**B**) Specificity of DL model in five algorithms. (**C**) AUROC of DL model in five algorithms. RNN: recurrent neural network; LSTM: long short-term memory; GRU: gated recurrent unit; Bi-LSTM: bidirectional long short-term memory; AUROC: area under the receiver operating characteristic; DL: deep learning.

**Figure 5 viruses-15-00304-f005:**
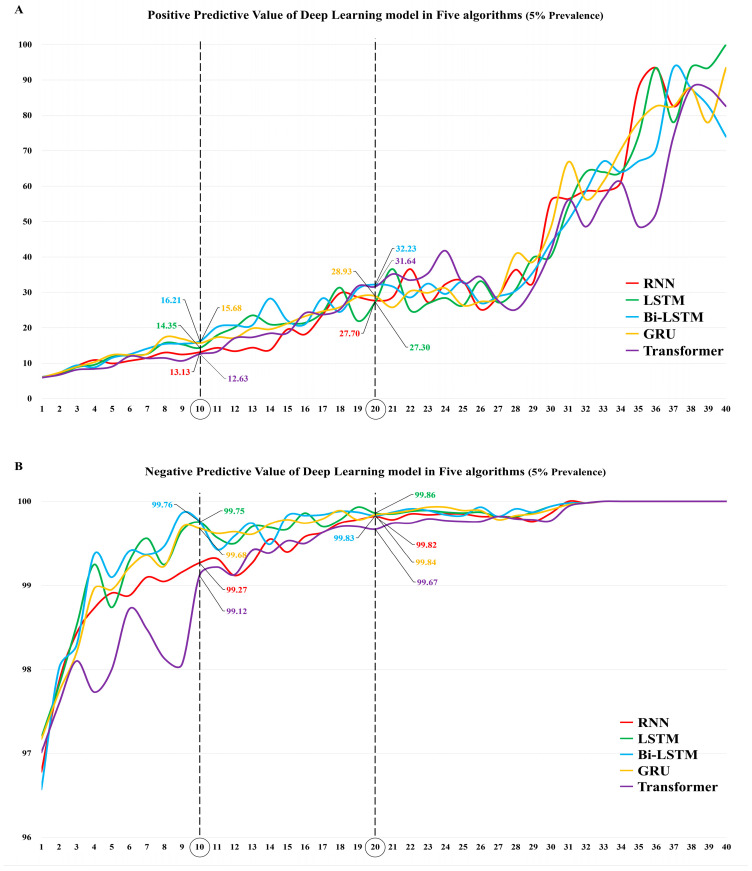
Diagnostic performance affected by the prevalence of 5% of DL models in five different algorithms using the test dataset. (**A**) Positive predictive value of DL model in five algorithms. (**B**) Negative predictive value of DL model in five algorithms. (**C**) Accuracy of DL model in five algorithms. RNN: recurrent neural network; LSTM: long short-term memory; GRU: gated recurrent unit; Bi-LSTM: bidirectional long short-term memory; DL: deep learning.

**Figure 6 viruses-15-00304-f006:**
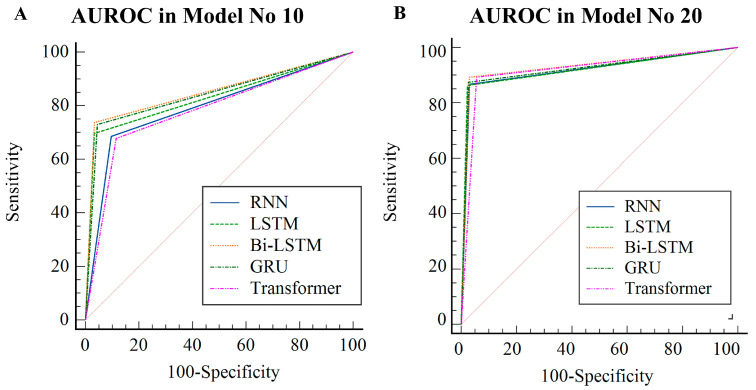
Differences in the AUROC of models 10 and 20 between the five different algorithms using the test dataset. (**A**) AUROC in Model No 10. (**B**) AUROC in Model No 10. RNN, recurrent neural network; LSTM, long short-term memory; GRU, gated recurrent unit; Bi-LSTM, bidirectional long short-term memory; AUROC, area under the receiver operating characteristic.

**Table 1 viruses-15-00304-t001:** Details on the hyper-parameters of DL models.

Hyper-Parameters		Optimal Value	Experimental Categories
Common	Loss function	Cross-entropy	-
	Optimization algorithm	Adam	SGD, Adam, momentum
	Learning rate	1 × 10^−3^	1 × 10^−2^–1 × 10^−6^
	Batch size	128	32–128
RNN-based approach(RNN, LSTM, Bi-LSTM, GRU)	No. of hidden units	128	32–256
	No. of hidden units in dense layer	32	32–128
	Dropout rate	0.3	0.3–0.6
Transformer	No. of heads	4	2–6
	No. of transformer layers	4	2–4
	No. of hidden units in multi-head self-attention	256	64–512
	No. of hidden units in dense layer	128	32–128
	Dropout rate	0.25	0.2–0.6

SGD: Stochastic Gradient Descent; RNN: recurrent neural network; LSTM: long short-term memory; BiLSTM: bidirectional long short-term memory; GRU: gated recurrent unit.

**Table 2 viruses-15-00304-t002:** Pairwise comparison of AUROC of Model 10 between 5 algorithms using test dataset.

Model 10	AUROC (95% CI)	Pairwise Comparison	*p* Value	Pairwise Comparison	*p* Value
RNN	79.44 (76.12–82.24)	RNN vs. LSTM	0.0036	LSTM vs. GRU	0.2372
LSTM	83.15 (80.25–85.92)	RNN vs. Bi-LSTM	<0.0001	LSTM vs. Transformer	0.0005
Bi-LSTM	85.19 (82.22–88.12)	RNN vs. GRU	0.0003	Bi-LSTM vs. GRU	0.2255
GRU	84.26 (81.56–87.12)	RNN vs. Transformer	0.2626	Bi-LSTM vs. Transformer	<0.0001
Transformer	78.15 (74.88–81.70)	LSTM vs. Bi-LSTM	0.0154	GRU vs. Transformer	<0.0001

AUROC: area under the receiver operating characteristics; CI: confidence interval; RNN: recurrent neural network; LSTM: long short-term memory; Bi-LSTM: bi-directional long short-term memory; GRU: gated recurrent unit.

**Table 3 viruses-15-00304-t003:** Pairwise comparison of AUROC of Model 20 between 5 algorithms.

Model 20	AUROC (95% CI)	Pairwise Comparison	*p* Value	Pairwise Comparison	*p* Value
RNN	91.85 (89.68–94.04)	RNN vs. LSTM	0.8275	LSTM vs. GRU	0.5638
LSTM	92.04 (89.86–94.19)	RNN vs. Bi-LSTM	0.1257	LSTM vs. transformer	0.8565
Bi-LSTM	93.15 (91.04–95.03)	RNN vs. GRU	0.5322	Bi-LSTM vs. GRU	0.3708
GRU	92.41 (90.30–94.48)	RNN vs. transformer	1.0000	Bi-LSTM vs. transformer	0.1931
Transformer	91.85 (89.50–94.00)	LSTM vs. Bi-LSTM	0.1313	GRU vs. transformer	0.6004

AUROC: area under the receiver operating characteristics; CI: confidence interval; RNN: recurrent neural network; LSTM: long short-term memory; Bi-LSTM: bi-directional long short-term memory; GRU: gated recurrent unit.

## Data Availability

All of the data generated or analysed during this study are included in this published article (and its Supplementary Information Files). If you need more details, the datasets generated during the current study are available from the corresponding author on reasonable request.

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
