# Peer review of "Comparison of the Diagnostic Performance of Deep Learning Algorithms for Reducing the Time Required for COVID-19 RT–PCR Testing"

_viruses, 2023, doi:10.3390/v15020304_

Round 1

Reviewer 1 Report

The authors propose a machine learning approach. based on five deep learning architecture, in order to predict the outcome of COVID-19 RT-PCR testing in a shorter time. They used RNN, LSTM, Bi-LSTM, GRU and Transformer as deep learning algorithm for the analysis of the fluorescence values obtained for a total of 40 cycles via the RT-PCR tests. Experiments showed report the result obtained after 10 and 20 cycles. Prediction results are measured through several metric scores, and the results of each algorithm are compared and discussed.
The paper is clear and well written, English is good.  The technical part of the paper is sound and results are good.
In my opinion, the following issues should be addressed:

Major remarks:
1- Similar approaches to your work, like the ones described from line 346 to 372, should be collected and briefly discussed in a “Related work” Section after Introduction
2- The deep learning architecture should be better described, including the MLP part of the models. Furthermore, you should talk about the parametrization of the networks and, to this aim, please provide a table summarizing the values of the main parameters, such as number of layers, layers’ size, optimizer, learning rate.
3- Did you perform cross-validation on your training set? Please clarify it.
4- On line 161-162, please clearly explain why you only considered models 10 and 20 in your analysis.
5- Are the results shown in the paper, either figures and tables, referred to validation set or test set? Please clarify and be sure to show both results.
6- Please provide datasets and source code, preferibly on a public git repository.

Minor remarks:
1- Please provide bibliographic reference for the five deep learning architecture.
2- Please provide details about the adopted deep learning framework (keras?, pytorch?)

Author Response

Thank you for your careful review and comments. Please see the attachment. 

Reviewer 2 Report

Thank you for sharing your finding and presenting them in this manuscript. I have some comments below.

1. The time series DL models should be introduced in the introduction.

2. Materials section, Figures of materials used in this study should be added.

3. In Sections 3.1 and 3.2, results should be arranged in tables.

4. Software and hardware platform for the DL models developed and deployed should be given.

5. DL model tuning strategy should be stated.

6. Axes' Labels in Figures 3 and 4 are too small.

7. Performance of the Transformer is the worst. It seems data size affected its performance. The performance variation among DL used should be discussed.

Author Response

(The authors gave the same response as above.)

Round 2

Reviewer 1 Report

Authors succesfully answered all my remarks

Reviewer 2 Report

Thanks for the update.